# Genome-Wide Identification and Expression Analysis of *CAMTA* Gene Family Implies *PbrCAMTA2* Involved in Fruit Softening in Pear

**Jinshan Yu** [1],[†], **Bobo Song** [1],[†], **Kaidi Gu** [2], **Beibei Cao** [1], **Kejiao Zhao** [1], **Jun Wu** [1],*** and **Jiaming Li** [1],***

[1]  State Key Laboratory of Crop Genetics & Germplasm Enhancement and Utilization,
    Nanjing Agricultural University, Nanjing 210095, China
[2]  National Key Laboratory of Crop Biology, College of Horticulture Science and Engineering,
    Shandong Agricultural University, Tai'an 271018, China
*   Correspondence: wujun@njau.edu.cn (J.W.); lijiaming@njau.edu.cn (J.L.); Tel.: +86-25-84396485 (J.W. & J.L.)
†   These authors contributed equally to this work.

**Abstract:** CAMTA are calcium-modulating binding transcription factors that contribute to plant development. We identified 46 *CAMTA* genes from eight Rosaceae species and divided them into five subgroups based on a phylogenetic tree. Our analysis indicated that CAMTA is a highly conserved family among Rosaceae species, with a conserved DNA-binding domain (CG-1) and a conserved transcription factor immunoglobulin domain (TIG). Following a recent whole-genome duplication event, the genomes of Chinese white pear, European pear, and apple experienced significant expansion, resulting in the number of *CAMTA* genes being twice that of the other species. Cis-element identification showed that the distribution of the zein metabolism regulation-responsive element was different in the promoters of Chinese white pear (55.56%) and European pear (11.11%) *CAMTA* gene families. The gene expression results showed that *PbrCAMTA1*, *2*, *6*, *7* was highly expressed in pear fruit. Among them, *PbrCAMTA2* may have a key influence on fruit softening, as observed in transient transformation experiments. In conclusion, our results provide crucial insights into the evolution of the *CAMTA* gene family in pear and other Rosaceae species and identify a candidate *PbrCAMTA* gene, which is involved in the dynamic development of pear fruits.

**Keywords:** pear; CAMTA; synteny analysis; qRT-PCR; fruit softening

## 1. Introduction

As the third most significant perennial fruit species, pears provide nutrients, antioxidants, and dietary fiber and are commonly used in traditional Chinese medicine. Pears can be divided into two groups: European pears and Asian pears [1]. After harvesting, most European pear species' flesh becomes soft quickly, and storability is poor. Some early ripening pear species have even shorter post-harvest storage periods. We know fruit quality and post-harvest storage capabilities are significant factors in the commercial quality assessment of fruit products, while fruit maturity determines the quality of fruit post-harvest and directly affects the current market and consumer choices [2,3]. Fruit softening manifests as a loss of fruit firmness, which is thought to be due to ethylene-mediated enzymatic modification of the cell wall, such as β-galactosidase (TBG), xyloglucan endotransglucosylase (XET), and Polygalacturonase (PG) [4,5]. Pectin galactose is hydrolyzed into glucose and galactose by β-galactosidase, and previous studies have reported that when loss of firmness is detected during persimmon fruits ripening, β-galactosidase activity increases almost 3.7-fold [6]. XET is associated with the swelling and loosening of the cell wall and shows an increase in activity during ethylene-induced softening in kiwifruit [4]. 1-Aminocyclopropane-1-Carboxylate Oxidase (ACO) participates in ethylene de novo synthesis, and *ACO1* is reported to have low expression levels in green tomatoes,

with its level increasing rapidly when the fruit begins to ripen [7]. Some transcription factors are also known to affect fruit ripening, such as calmodulin-binding transcription activator (CAMTA), which is extremely responsive to ethylene and regulates tomato fruit ripening [8].

The CAMTA family is a highly conserved gene family in plants. Since it was first discovered in tobacco [9], the *CAMTA* gene family has been identified in many other plants, such as *Arabidopsis thaliana* [10], *Solanum lycopersicum* [8], *Vitis vinifera* [11], *Zea mays* [12] *Fragaria ananassa* [13], *Populus trichocarpa* [14], *Gossypium* [15], *Citrus sinensis* [16], *Musa acuminata* [17], and *Linum usitatissimum* [18]. CAMTA plays many significant roles in plants, and it has been reported to have a wide range of functions. CAMTA has been discovered in ethylene-induced conditions [19]: *NtER1/CAMTA* increases with ethylene induction in tobacco, and the content of CAMTA is significantly higher in senescent leaves than in young leaves [9]. *AtSR1/AtCAMTA3* directly regulates NDR1 and EIN3 and is involved in the plant ethylene-related aging process [20]. After treating tomato with exogenous ethylene, there is a significant increase in the expression of seven *SlCAMTA/SR* genes in tomatoes, suggesting that *CAMTA* might act as an ethylene responder to regulate fruit ripening [8]. In addition, the expressions of *PG*, *PE1*, *CEL2*, and other fruit wall metabolic genes are promoted in *SlSR4* mutants, suggesting that *CAMTAs* are involved in the tomato fruit softening process. *CAMTA* genes respond to different biotic and abiotic stresses. *AtCAMTA3* is involved in the regulation of the cold response of *Arabidopsis* by positively regulating the expression of *CBF2* and improving cold resistance [21]. *CAMTA* genes mediate SA biosynthesis in *Arabidopsis thaliana* [22]. These studies have proved that the CAMTA family has major effects on plant growth, fruit maturation, and stress regulation.

Despite systematic functional research related to *CAMTA* genes being extensive in a number of different plants, the *PbrCAMTA* gene family in pear has not yet been researched. Thanks to the publication of the genome sequence of *P. bretschneideri* [23], it is now possible to carry out the relevant research. Here, we used eight Rosaceae genomes, *Pyrus bretschneideri* (Chinese white pear), *Pyrus communis* (European pear), *Fragaria vesca* (strawberry), *Prunus mume* (Japanese apricot), *Prunus persica* (peach), *Rubus occidentalis* (black raspberry), *Malus domestica* (apple), *Prunus avium* (sweet cherry), and two identified model plants (*Arabidopsis thaliana* [10] and Tomato [8])to cluster and analyze the CAMTA protein family. We performed analysis on genome-wide identification, phylogenetic relationship, chromosome distribution, genomic structure, expression patterns, and function verification of *PbrCAMTA* genes. Our findings have the potential to lead to a better understanding of the roles and structures of *CAMTA* genes in Rosaceae species, especially pear, and can be useful for screening candidate genes that may be associated with fruit softening, this way laying the foundation for function identification.

## 2. Materials and Methods

### 2.1. Whole-Genome Identification of CAMTA Genes

The genome data of eight Rosaceae species were downloaded via the GDR database (https://www.rosaceae.org/, accessed on 5 December 2021). We then used two different methods to identify the genes of the *CAMTA* family. Firstly, the Pfam file of the CAMTA domain was used to identify candidate *CAMTA* genes in eight Rosaceae species (Chinese white pear, European pear, Strawberry, Japanese apricot, peach, black raspberry, apple, and sweet cherry) using HMMSearch software. The protein sequences of the CAMTA family of *Arabidopsis* and tomato were subsequently downloaded [8,10]. Using the CAMTA protein sequences in *Arabidopsis* and tomato as a query, we identified candidate *CAMTA* genes in the genomes of eight Rosaceae species using the blastp software. We used the overlapping results of both methods for further analysis. We employed the CDD tool (https://www.ncbi.nlm.nih.gov/cdd/, accessed on 10 December 2021), SMART tool (http://smart.emblheidelberg.de/, accessed on 10 December 2021), and InterProScan tool (http://www.ebi.ac.uk/Tools/pfa/iprscan/, accessed on 10 December 2021) to verify the completeness of the CAMTA domain and to perform a functional analysis of the correspond-

ing protein sequence. In addition, we used the Expasy tool (https://www.expasy.org/, accessed on 5 December 2021) to predict protein molecular weights (Mw) and isoelectric points (PI) for the eight species considered.

### 2.2. Multiple Sequence Alignment and Phylogenetic Tree Analysis

We utilized ClustalW with the default parameters to align multiple protein sequences in ten species (eight Rosaceae species, *Arabidopsis*, and tomato) and then used the Genedoc software to visualize the results of multiple sequence alignment. Using the multi-sequence alignment result, the neighbor-joining method (NJ) was used, 1000 bootstraps were set, and an unrooted phylogenetic tree was constructed with MEGA-X software.

### 2.3. Gene Structure Analysis and Conserved Motifs Analysis

We obtained *CAMTA* gene structure information and chromosome position information of the *CAMTA* genes of eight Rosaceae species from genome databases using our in-house Python scripts. The gene structure of the *CAMTA* genes in Rosaceae species was visualized with the TBtools tool [24]. The MEME tool (Multiple Em for Motif Elicitation) (http://meme-suite.org/tools/meme/, accessed on 30 December 2021) was used to identify conserved motifs, and the parameters were set as follows: repetitions, any number; the number of different motifs, 20; minimum motif width, 30; and maximum motif width, 70. We obtained merged visualizations through the iTOL tool.

### 2.4. Chromosomal Location and Synteny Analysis

Chromosome location information of *CAMTA* genes was visualized with TBtools [24]. Homologous gene pairs were identified using the MCScanX software, and the circle picture was generated by the Circos software (version 0.69.2).

### 2.5. Cis-Regulatory Elements Analysis of Putative Promoters

All promoter sequences of *CAMTA* genes were extracted using the "getfasta" function in Bedtools to extract 2000 bp sequences upstream of the transcriptional initiation sites (putative promoter regions). We then used PlantCARE (http://bioinformatics.psb.ugent.be/webtools/plantcare/html/, accessed on 18 January 2022) to predict cis-regulatory elements of the *PbrCAMTA* gene promoter regions.

### 2.6. Transcriptome Expression Pattern Analysis

We used the transcriptome data from six fruit developmental periods of 'Dangshansuli' cultivar (15 DAFB (days after full bloom), 36 DAFB, 80 DAFB, 110 DAFB, 145 DAFB, and 167 DAFB, which correspond, respectively to DS1, DS2, DS3, DS4, DS5, DS6 in 'Dangshansuli'), the transcriptome data from seven fruit developmental periods of other five pear cultivar ('Hosui', 'Kuerlexiangli', 'Nanguoli', 'Starkrimson' and 'Yali'), and the transcriptome data from six tissues (leaf, fruit, petal, sepal, ovary and stem) of 'Dangshansuli' cultivar. These samples were obtained from previous studies of our research group [25–27]. Based on the RPKM value (Reads Per Kilobase Per Million Mapped Reads) of the *PbrCAMTA* gene, $\log_2$(RPKM+0.01) was calculated, and the heatmap was drawn using the "pheatmap" function in R.

### 2.7. Quantitative Real-Time PCR Analysis (qRT-PCR)

We collected samples from six tissues (leaf, fruit, petal, sepal, ovary, and stem) and fruit tissues at six developmental periods (15 DAFB, 36 DAFB, 80 DAFB, 110 DAFB, 145 DAFB and 167 DAFB) of 'Dangshansuli' in order to perform qRT-PCR analysis to validate the results of transcriptome data. The RNA was extracted and reverse transcribed to synthesize cDNA (TransGen Biotech Co. Ltd., Beijing, China), primers were designed using the NCBI website (https://www.ncbi.nlm.nih.gov/tools/primerblast, accessed on 8 December 2021), and the corresponding candidate gene sequences were amplified. qRT-PCR analysis was performed using LightCycler 480 SYBR GREEN I Master (Roche, Beijing, China). The

analysis was conducted using three biological and three technical repeats. The $2^{-\Delta\Delta Ct}$ method was used to calculate the relative gene expression levels in different samples, and all of the data were normalized based on stem or 15 DAFB samples.

### 2.8. Transient Transformation of Pear Fruits

To transiently overexpress *PbrCAMTAs*, we constructed the p1300-35S:PbrCAMTA2-GFP vector and transferred it into *Agrobacterium tumefaciens* strain GV3101 using the freeze–thaw method. We centrifuged and collected the cells and then re-suspended them in an infiltration buffer mixed with 10 mM MgCl$_2$, 10 mM MES (pH 5.7), and 200 µM acetosyringone (AS) to OD$_{600}$ 0.8–1.0 and induced at 20–25 °C for 4 h. The cells were agro-infiltrated into 'Zaosu' pear fruits at 115 DAFB using 1 mL needleless syringes. We identified six injection locations along the equatorial line of each pear fruit so that the same fruit could be injected three times with *PbrCAMTA2* and three times with an empty vector. Five fruits were injected for each experiment. The injected pear fruits were placed overnight in the dark at 25 °C and incubated at 25 °C constant temperature for five days before being examined. Three biological replicates were taken. After measuring the firmness, the fruit tissue was collected 1–2 cm deep from the injection site and stored at −80 °C.

### 2.9. Firmness Measurement

A probe compressed the pear fruit tissues near the injection holes at an equatorial position at 4 mm and 1.5 mm/s speed to obtain the mean peak force using the Brookfield CT3 Texture Analyzer. The measurement was taken on three experiment-injected holes and three empty vector-injected holes in each fruit. Five pear fruits were used as biological replicates in the experiment, and the unit (kg/cm$^2$) was used.

## 3. Results

### 3.1. Genome-Wide Identification of the CAMTA Gene Family in Chinese White Pear and Other Rosaceae Species

We identified 46 *CAMTA* genes from eight Rosaceae species (nine in European pear, nine in Chinese white pear, eight in apple, five in peach, four in sweet cherry, four in Japanese apricot, four in black raspberry, and three in strawberry) (Figure 1). Chinese white pear, European pear, and apple had double the number of *CAMTA* genes than the other Rosaceae species. In order to distinguish members of the *CAMTA* gene family, we renamed all *CAMTA* genes and predicted the molecular weight (Mw) and isoelectric point (pI) of the CAMTA protein sequences with the ExPASy Proteomics Server (Supplementary File S6: Table S2). The length of the CAMTA-encoded protein sequences ranged from 854 (*PcpCAMTA2*) to 2065 (*FvH4CAMTA1*) amino acids, the protein mass ranged from 96 kD (*PcpCAMTA2*) to 233 kD (*FvH4CAMTA1*), and protein pIs ranged from 5.26 (*PavCAMTA3*) to 8.16 (*FvH4CAMTA3*). In Chinese white pear, the length of the CAMTA-encoded protein sequences ranged from 865 (*PbrCAMTA9*) to 1152 (*PbrCAMTA6*) amino acids, the protein mass ranged from 97 kD (*PbrCAMTA9*) to 128 kD (*PbrCAMTA6*), and protein pIs ranged from 5.38 (*PbrCAMTA7*) to 7.13 (*PbrCAMTA3*).

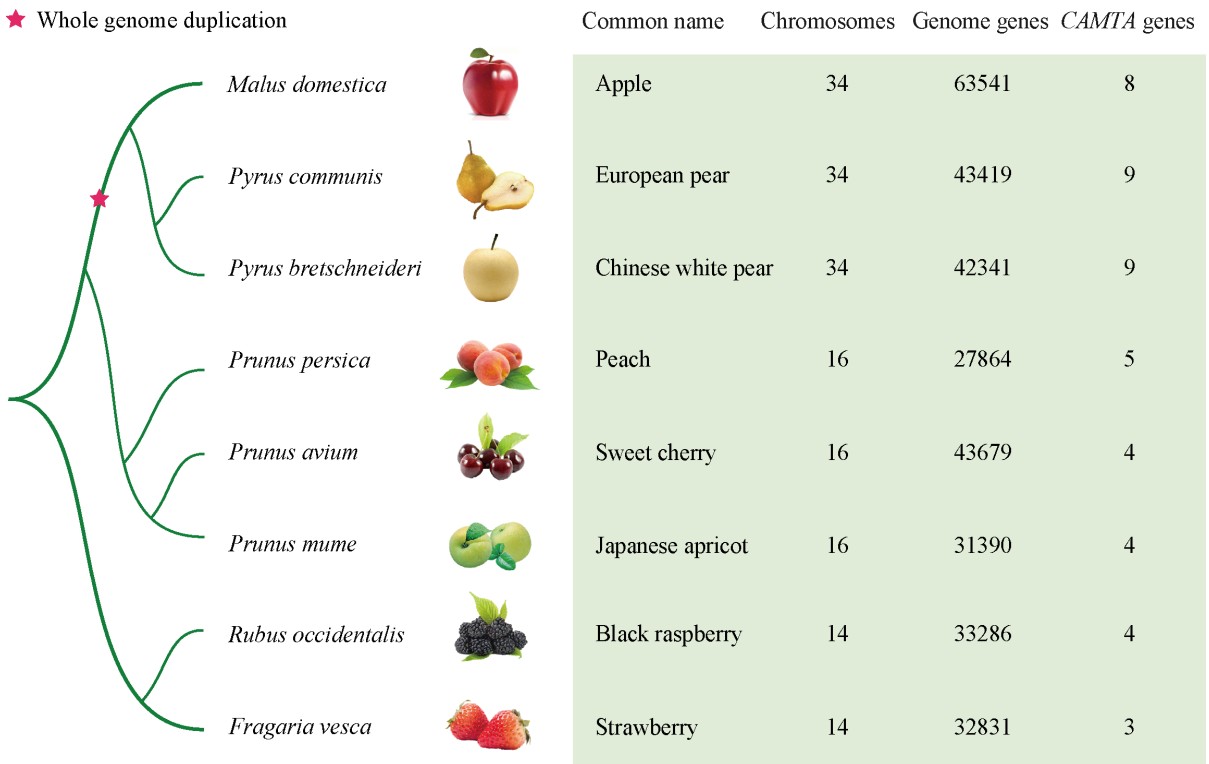

**Figure 1.** Phylogenetic tree and genome information of *CAMTA* genes in eight Rosaceae species. The pentagram indicates the occurrence of whole-genome duplication events (WGD).

### 3.2. Phylogenetic Analysis of CAMTA Protein Family in Pear and Other Rosaceae Species

A neighbor-joining (NJ) phylogenetic tree was constructed with the Mega-X program using the CAMTA proteins of *Arabidopsis*, tomato, and eight Rosaceae species (Figure 2). The total 59 CAMTA proteins of the ten species were subsequently divided into five subgroups: Group I–Group V. The distribution in the five subgroups was as follows: in European pear, one protein was in Group I, two in Group II, two in Group III, two in Group IV, and two in Group V; in Chinese white pear, two proteins were in Group I, two in Group II, one in Group III, two in Group IV, and two in Group V. The difference in the group distribution of the proteins observed between European and Chinese pear might be due to the divergence between the two species. In apple, one protein belonged to Group I, two to Group II, one to Group III, two to Group IV, and two to Group V. The distribution of CAMTA proteins of Japanese apricot and black raspberry was the same, with one protein in Group I, one in Group II, one in Group III, 0 in Group IV, and one in Group V. In strawberry, there was one protein in Group I, 0 in Group II, one in Group III, 0 in Group IV, and one in Group V; in peach, there was one in Group I, one in Group II, one in Group III, one in Group IV, and one in Group V; in sweet cherry, 0 in Group I, 1 in Group II, one in Group III, one in Group IV, and one in Group V (Supplementary File S7: Table S3). We found CAMTA proteins from different species clustered into the same subgroup, suggesting they may have similar functions.

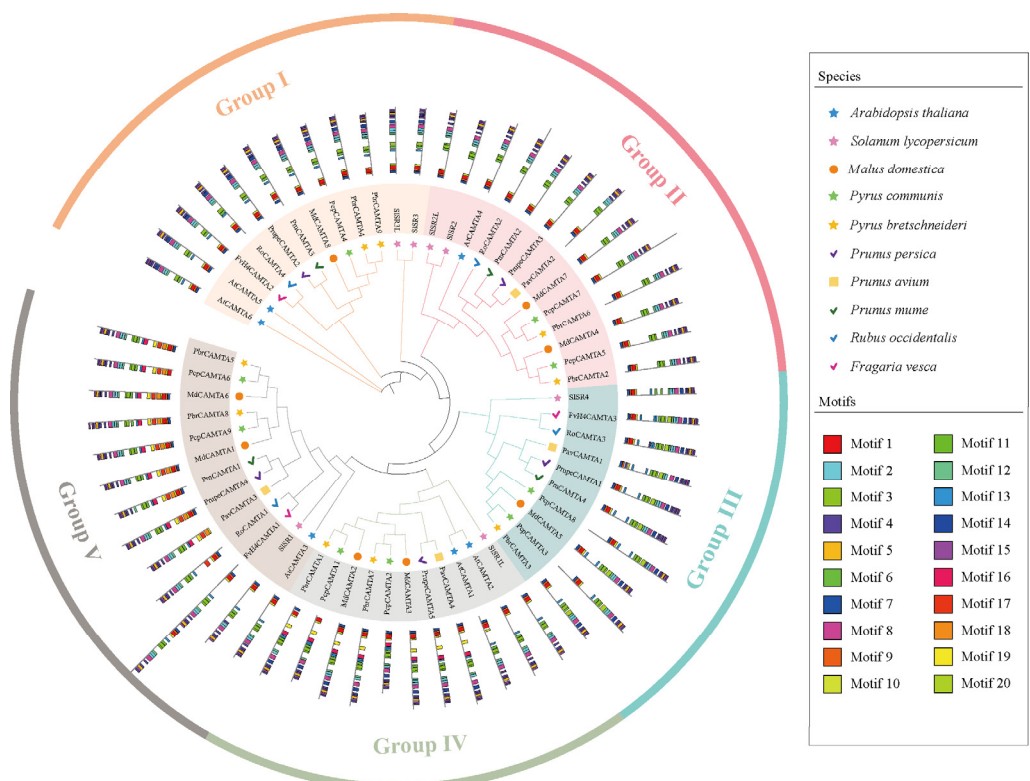

**Figure 2.** Phylogenetic tree and distribution of conserved motifs across the CAMTA protein family in eight Rosaceae species. The phylogenetic tree of the CAMTA protein family is constructed by iqtree using the neighbor-joining (NJ) method with 1000 bootstraps. A total of 20 motifs are predicted using the MEME tool. Among these, yellow branches indicate Group I; pink branches indicate Group II; blue branches indicate Group III; gray branches indicate Group IV; and brown branches indicate Group V. Blue star, yellow star, green star and pink star indicate the CAMTA proteins, respectively, in *Arabidopsis*, *P. bretschneideri*, *P. communis*, and *S. lycopersicum*. Pink tick, green tick, purple tick, and blue tick indicate CAMTA proteins, respectively, in *F. vesca*, *P. mume*, *P. persica*, and *R. occidentalis*. The orange circles and squares represent CAMTA proteins in *M. domestica* and *P. avium*.

### 3.3. Conserved Motif and Gene Structure Analysis of CAMTA Family in Rosaceae Species

We identified 20 conserved motifs in the CAMTA proteins among ten species and visualized their amino acid compositions (Supplementary File S1: Figure S1) using the MEME tool. The structures and locations of the conserved motifs in each subgroup were nearly identical, indicating that the proteins from the same subgroup likely have similar functions. Based on motif analysis and phylogenetic tree results, we observed that the domains of the CAMTA proteins in the eight Rosaceae species were relatively conserved; in fact, all proteins except PavCAMTA3 contained Motif 1, and all proteins had Motif 3. As confirmed by the results of multi-sequence comparisons (Supplementary File S2: Figure S2), Motif 1 (CG-1) and Motif 3 (TIG) were two highly conserved functional domains in all CAMTA proteins found in the eight Rosaceae species considered in our study. Interestingly, all proteins had a conserved domain consisting, in order, of Motif 4, Motif 5, Motif 15, and Motif 4, indicating that this may be a highly conserved functional domain in CAMTA. The composition of conserved motifs in the same subgroup was almost identical, such as in Group I, which did not contain Motif 11; Group II did not contain Motif 13; Group V contained Motif 18, which suggested that the proteins in the same subgroups may have similar structures and functions. Conserved motif analysis provided reliable evidence and strongly supported the results of the clustering in the phylogenetic tree.

In order to clarify *CAMTA* gene structures and explore their structural diversity, we analyzed the intron and exon components by aligning their genomic sequences with

the CDS of eight Rosaceae *CAMTA* genes. As shown in Figure 3, the exon number of the 46 *CAMTA* genes ranged from 10 (*PavCAMTA3*, sweet cherry) to 24 (*FvCAMTA1*, strawberry), suggesting that the *CAMTA* gene structures in the Rosaceae family are diverse.

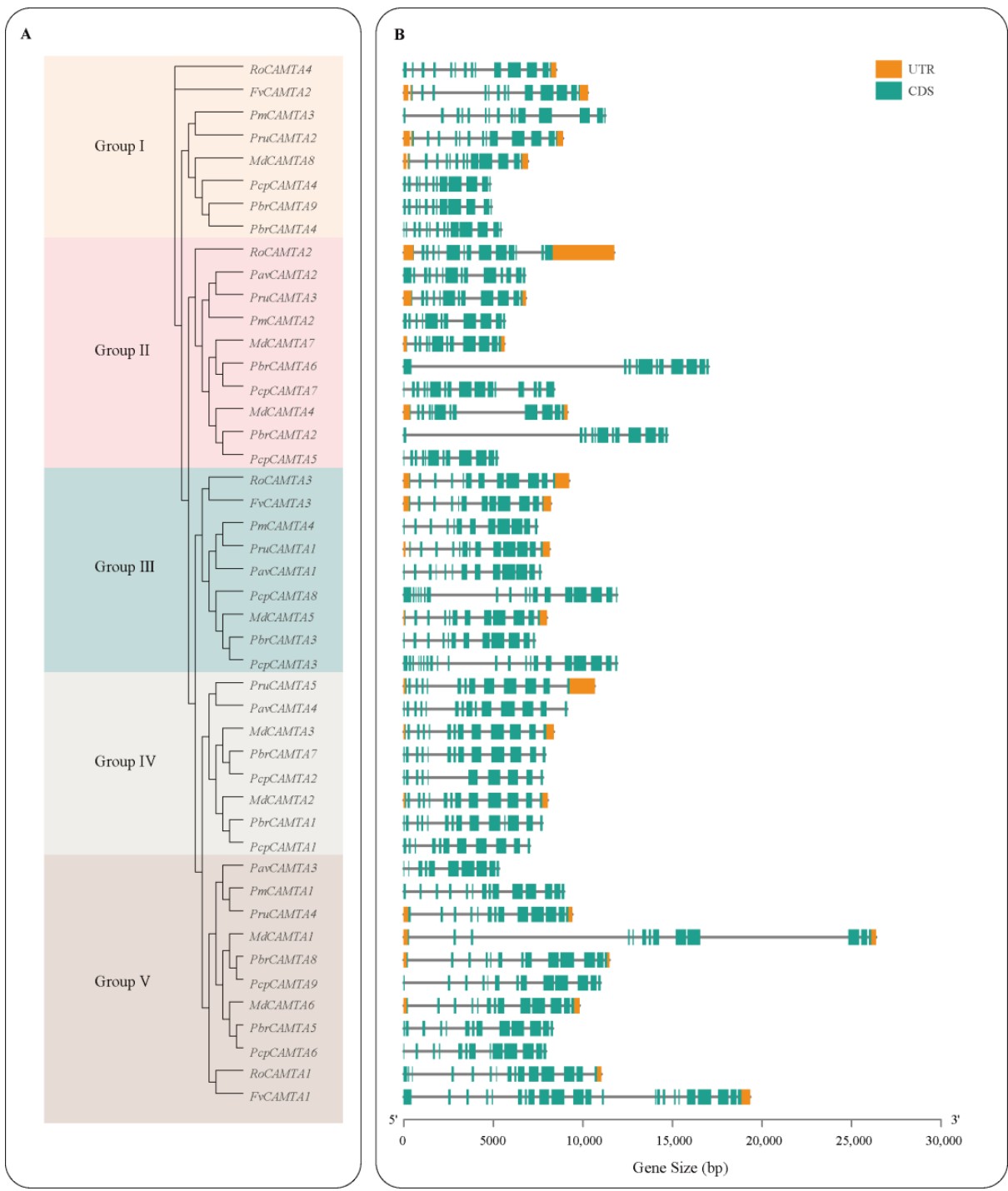

**Figure 3.** Phylogenetic relationship and gene structure of the *CAMTA* gene family in eight Rosaceae species. (**A**) The phylogenetic tree is constructed by iqtree using the neighbor-joining (NJ) method with 1000 bootstraps. (**B**) Intron–exon structures of *CAMTA* genes in eight species.

### 3.4. Chromosomal Distribution and Synteny Analysis of CAMTA Genes in Pear and Other Rosaceae Species

We analyzed the genomic distribution of *CAMTA* genes on the chromosomes of eight species. Six of the nine *PbrCAMTA* genes in Chinese white pear were unevenly distributed throughout four chromosomes (Figure 4A). *PbrCAMTA1* was on chromosome 5, while *PbrCAMTA2*, *PbrCAMTA3*, and *PbrCAMTA4* were co-located on chromosome 13. *PbrCAMTA5* was on chromosome 15, and *PbrCAMTA6* was on chromosome 16. The remaining three *PbrCAMTA* genes were mapped to different scaffold contigs. We analyzed the collinearity relationships of *CAMTA* genes between Chinese white pear and the other seven Rosaceae species in order to better understand the evolutionary mechanism of *CAMTA* genes. As shown in Figure 4A, we observed good collinearity in that nine *PbrCAMTA* genes in Chinese white pear separately corresponded to 14 homologous gene pairs in apple, nine pairs in European pear, ten pairs in peach, five pairs in sweet cherry, eight pairs in Japanese apricot, six pairs in black raspberry, and four pairs in strawberry. There were more collinear gene pairs between Chinese white pear and European pear or Chinese white pear and apple, indicating that these three species have close relationships, which is consistent with our analysis. Here, we identified four homologous *PbrCAMTA* gene pairs (Figure 4B), which contained eight *PbrCAMTA* genes (*PbrCAMTA1-PbrCAMTA7*, *PbrCAMTA2-PbrCAMTA6*, *PbrCAMTA4-PbrCAMTA9* and *PbrCAMTA5-PbrCAMTA8*), while *PbrCAMTA3* was the only gene with no collinear gene pair. Through further duplication event type analysis, we found that eight *PbrCAMTA* genes all originated from a WGD/segmental duplication event, while *PbrCAMTA3* originated from a dispersed duplication event (Supplementary File S8: Table S4).

### 3.5. Cis-Acting Elements Analysis in the Putative Promoter of CAMTA Genes in Rosaceae Species

The function and expression of a gene are related to the type of cis-acting element contained in the upstream promoters. We identified a total of 6584 cis-acting elements for 45 *CAMTA* gene promoters in the eight Rosaceae species considered. We selected and clustered 12 varieties of important cis-acting elements for further analysis (Figure 5, Supplementary File S9: Table S5): Abscisic acid (ABA)-responsive elements, Anaerobic-induction-responsive elements, Defense-and-stress-responsive elements, Gibberellins (GA)-responsive elements, Light-responsive elements, Methyl Jasmonate (MeJA)-responsive elements, Salicylic acid (SA)-responsive elements, MYBHv1 binding sites, Zein-metabolism-regulation-responsive elements, Low-temperature-responsive elements, and Meristem expression regulatory elements. All Rosaceae *CAMTA* had between one (*PavCAMTA3*) and twenty-two (*RoCAMTA3*) light-responsive elements, and 77.78% of the genes of all *CAMTAs* had ABA-responsive elements. Interestingly, we noticed that 55.56% of the *PbrCAMTA* genes of Chinese white pear had the zein metabolism-regulation-responsive element, while only 11.11% of the *PcpCAMTA* genes of European pear had that element, leading to divergence in the two pear species. The zein metabolism-regulation-responsive element is reported to be associated with fruit softening [28]. Our findings indicate that this element may have an effect on gene expression and may be involved in fruit maturity phenotypes in Chinese white pear and European pear.

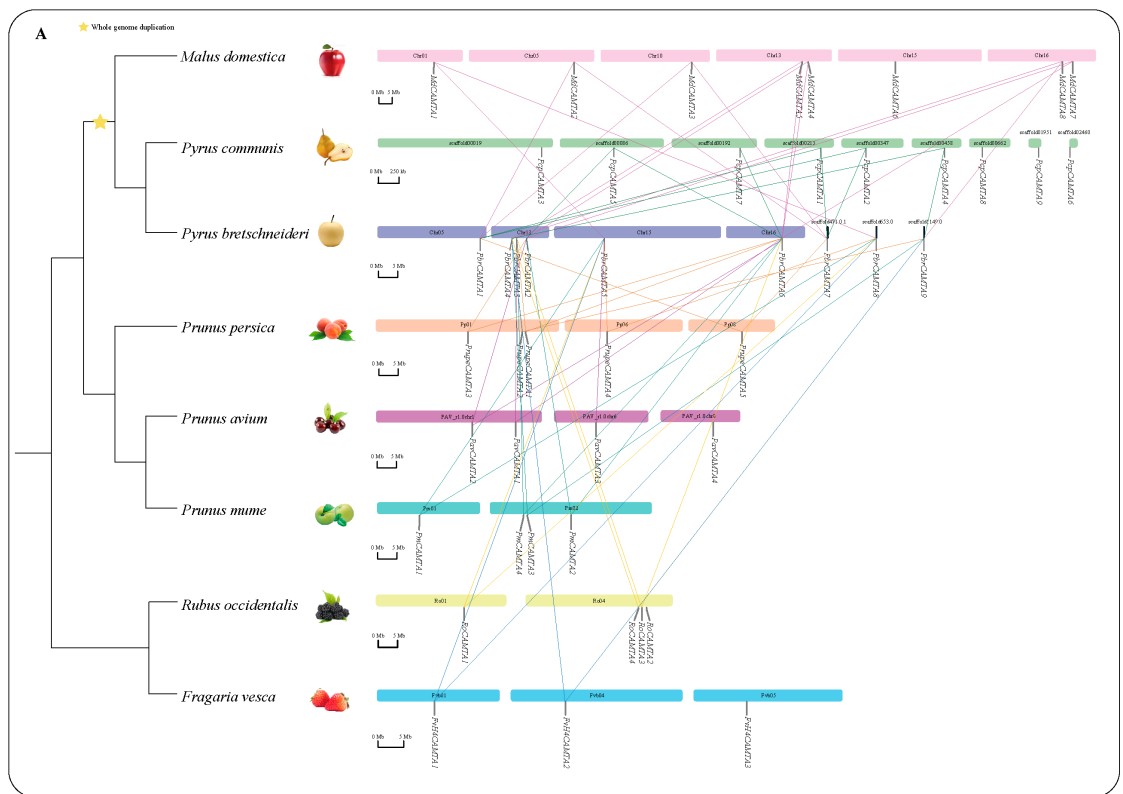

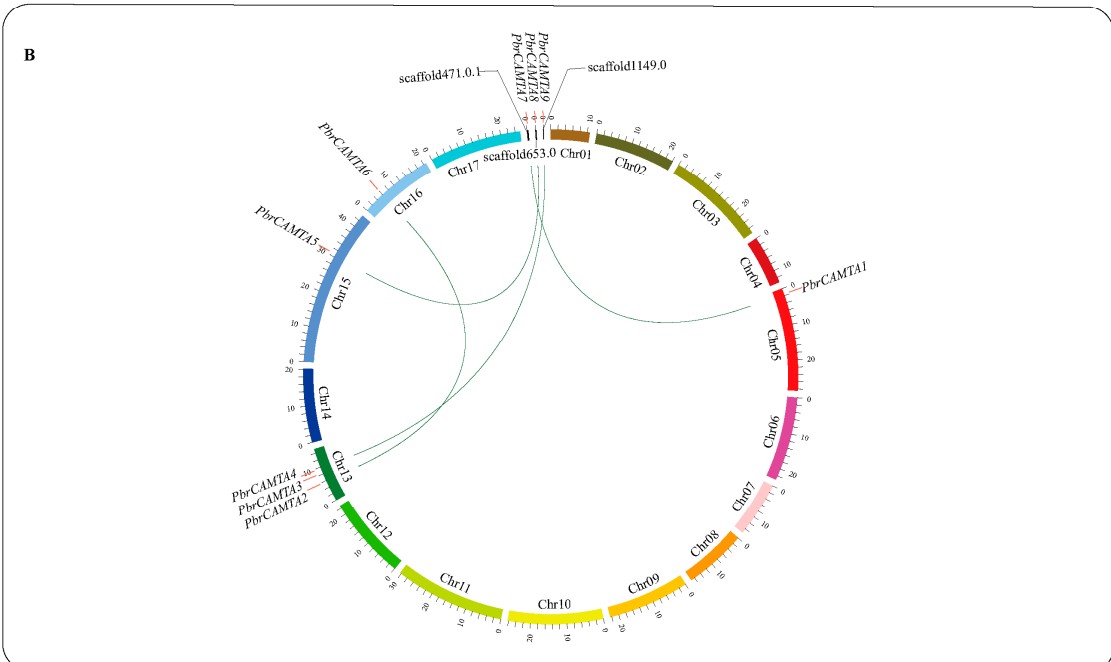

**Figure 4.** Synteny analysis in Chinese white pear. (**A**) Chromosomal distribution and synteny analysis of pear and seven other species. The chromosomes of different species are shown in different color blocks. Short lines located on different blocks represent the approximate distribution of each *CAMTA* gene. Long lines denote the details of collinear gene pairs between Chinese white pear and the other seven species. (**B**) Gene location and synteny analysis of *PbrCAMTA* genes. The homologous genes of the *PbrCAMTA* family are identified using MCScanX software, and the circle picture was plotted using Circos software (version 0.69.2). Red short lines indicate the position distribution of *PbrCAMTA* genes on 17 pear chromosomes. Green curves show the collinear gene pairs of the *PbrCAMTA* gene family.

### 3.6. Transcriptome and qRT-PCR Analysis of PbrCAMTA Genes

We analyzed the transcriptome data of six tissues (stem, leaf, petal, fruit, ovary, and sepal) of 'Dangshansuli' to obtain the expression patterns of *PbrCAMTAs*. The RPKM values represent the expression levels of *PbrCAMTAs*, and the heatmap result is shown in Figure 6A. We considered genes not to be expressed for RPKM values <= 1 and genes to have a low expression when 1 < RPKM < 3. *PbrCAMTA* genes were clustered into two classes based on the transcriptome data analysis: Class I included *PbrCAMTA1*, *2*, *6*, *7*; Class II included *PbrCAMTA3*, *4*, *5*, *8*, *9*. We found that the four genes (*PbrCAMTA1*, *2*, *6*, *7*) in Class I were highly expressed in all six tissues, meaning they may have different functions in different tissues, and they may play important roles in plant growth and development. On the contrary, Class II genes may function only in specific tissues, such as *PbrCAMTA5* and *PbrCAMTA8* in leaves, *PbrCAMTA3* in stems, and *PbrCAMTA9*, which was highly expressed in the petal tissue. In addition, we selected four *PbrCAMTA* genes (*PbrCAMTA2*, *4*, *7*, *8*) to verify their expression levels in six tissues of 'Dangshansuli' by quantitative RT-PCR experiment. As shown in Figure 6B, four genes had specific expression patterns, which were largely consistent with our transcriptome data demonstrating its reliability.

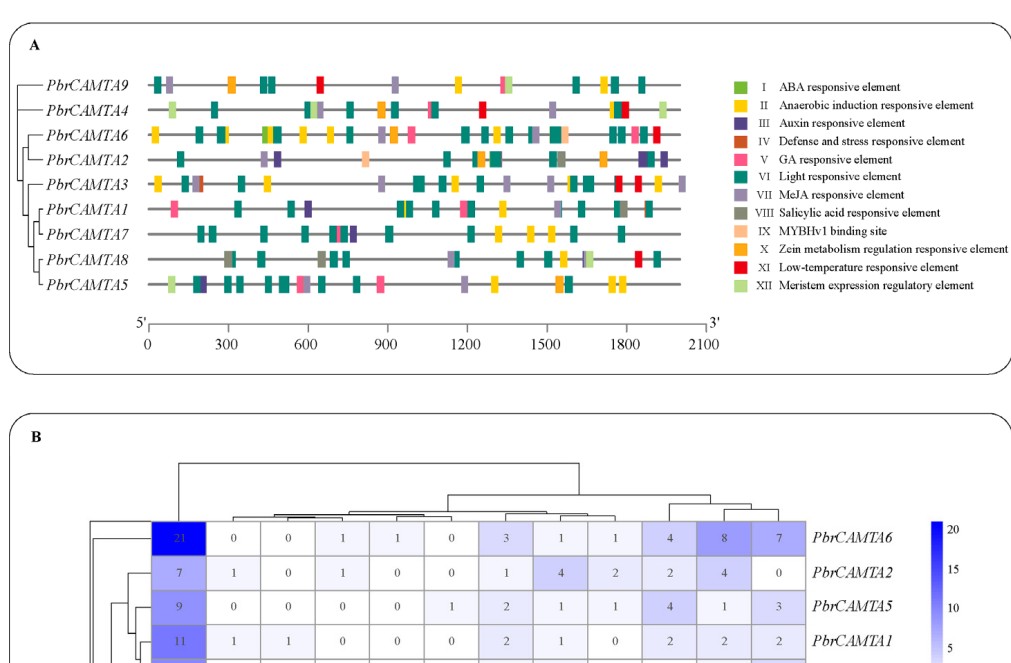

**Figure 5.** Cis-acting elements in putative promoters (2000 bp upstream) of *PbrCAMTA* genes in Chinese white pear. (**A**) Distribution of cis-acting elements on putative promoters of *PbrCAMTA* genes. The phylogenetic tree of the *PbrCAMTA* genes family was constructed by using iqtree using the neighbor-joining (NJ) method and 1000 bootstraps. (**B**) The number of cis-acting elements on putative promoters of *PbrCAMTA* genes. A total of twelve cis-acting elements are investigated, including: (I) ABA-responsive elements, (II) Anaerobic induction-responsive elements, (III) Auxin-responsive elements, (IV) Defense-and-stress-responsive elements, (V) GA-responsive elements, (VI) Light-responsive elements, (VII) MeJA-responsive elements, (VIII) Salicylic-acid-responsive elements, (IX) MYBHv1 binding sites, (X) Zein metabolism-regulation-responsive elements, (XI) Low-temperature-responsive elements, and (XII) Meristem expression regulatory elements.

In order to investigate the expression patterns of *PbrCAMTA* genes during pear fruit development in different pear cultivars, we analyzed the fruit transcriptome data of six major cultivars: 'Hosui', 'Kuerlexiangli', 'Nanguoli', 'Starkrimson', 'Yali' and 'Dangshansuli'. The results (Figure 7A) showed that nine *PbrCAMTA* genes had specific expression trends. Among these genes, *PbrCAMTA1*, *2*, *6*, and *7* were highly expressed in six periods of all cultivars; *PbrCAMTA3*, *4*, *5*, *8*, and *9* had relatively high levels of expression in some periods. The diversity of *PbrCAMTAs* expression patterns in fruits by qRT-PCR (Figure 7B) demonstrates that these genes may play different roles in fruit development. This observation lays a significant foundation for future research on gene functions.

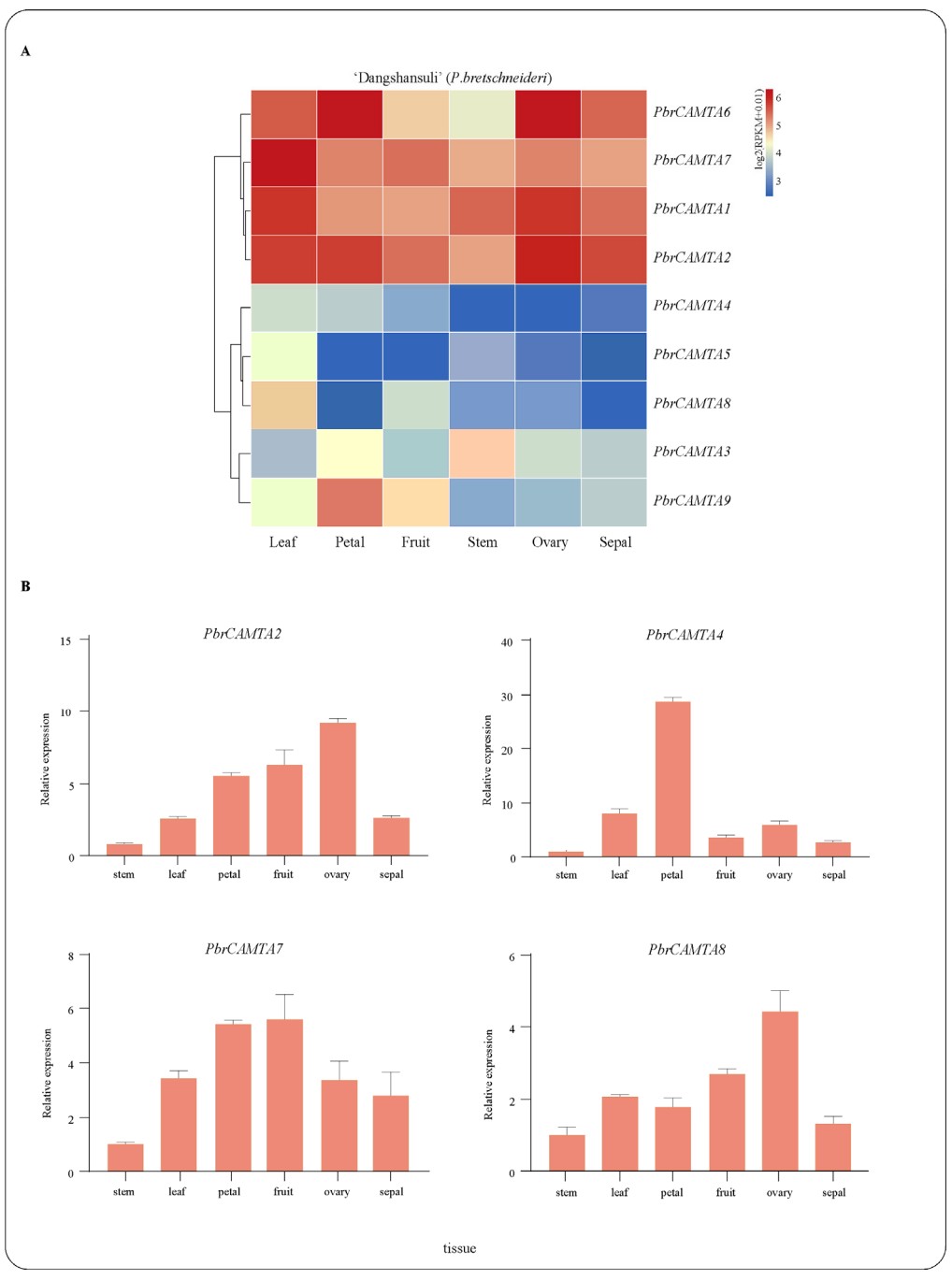

**Figure 6.** Heatmap and qRT-PCR analysis of expression levels of *PbrCAMTA* genes in tissues. (**A**) Heatmap of the expression level ($\log_2$(RPKM+0.01)) of nine *PbrCAMTA* genes in six different tissues (leaf, petal, fruit, stem, ovary, and sepal) in 'Dangshansuli'. The heatmap is plotted by pheatmap

red indicates high expression, and blue indicates low expression. The color scale at the top right represents RPKM values normalized by log2. (**B**) qRT-PCR analysis of four *PbrCAMTA* genes in six different tissues. The x-axis represents six different tissues (stem, ovary, petal, sepal, fruit, and leaf), and the y-axis represents relative expression levels of the *PbrCAMTA* genes. Error bars indicate three technical replicates derived from one bulked biological replicate.

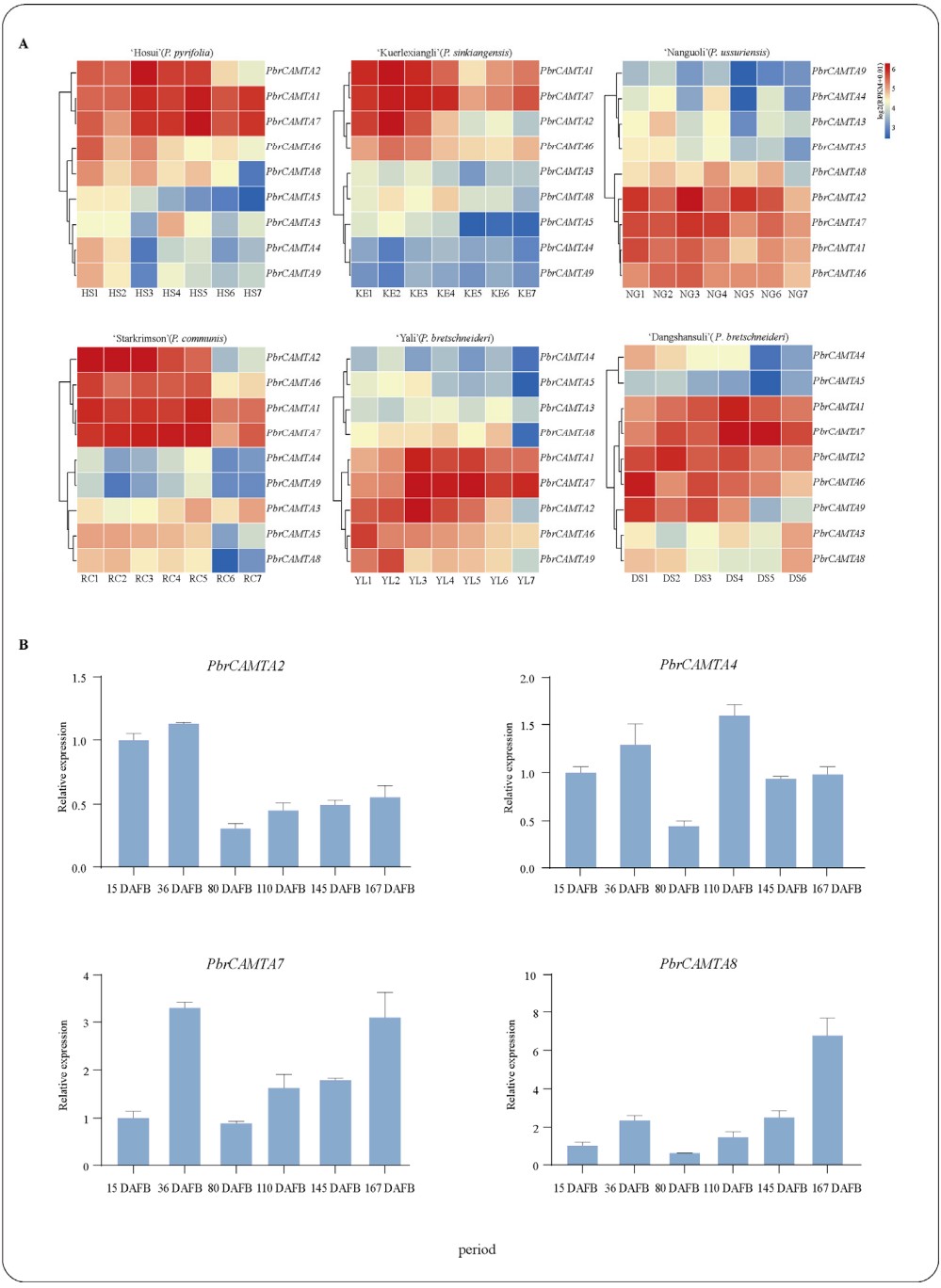

**Figure 7.** Heatmap and qRT-PCR analysis of expression levels of *PbrCAMTA* genes in different periods of six pear cultivars. (**A**) Heatmaps of expression level analysis of *PbrCAMTA* genes at DAFB of six different pear cultivars ('Hosui', 'Kuerlexiangli', 'Nanguoli', 'Starkrimson', and 'Yali'). (**B**) qRT-PCR analysis of four *PbrCAMTA* genes (*PbrCAMTA2*, *PbrCAMTA4*, *PbrCAMTA7*, and *PbrCAMTA8*) in fruit tissues of 'Dangshansuli' at different DAFB. The x-axis represents six different tissues, including stem, ovary, petal, sepal, fruit, and leaf, and the y-axis represents relative expression levels of *PbrCAMTA* genes. Error bars indicate three technical replicates derived from one bulked biological replicate.

### 3.7. Effects of PbrCAMTA Expression Manipulation

Based on transcriptome data and analysis, we hypothesized that *PbrCAMTA2* could be related to fruit ripening. To validate the function of *PbrCAMTA2*, we transiently expressed *PbrCAMTA2* in 'Zaosu' pear fruits at 110 DAFB by injecting *Agrobacterium* containing the p1300-35S:PbrCAMTA2-GFP construct vector. We injected the empty vector as a control at the symmetrical position in the treated pear fruits (Figure 8A). Seven days after the injection, the pear fruit tissues were collected to measure the firmness and extract the RNA. We found that the tissues that overexpressed *PbrCAMTA2* had better firmness than those injected with the empty vectors (Figure 8B). A quantitative RT-PCR experiment was used to investigate *PbrCAMTA2* and the other genes involved in fruit ripening, including, xyloglucan endotransglucosylase (*XET*) [29], and 1-Aminocyclopropane-1-Carboxylate Oxidase (*ACO*) [30] and β-galactosidase (*TBG*) [31] The results showed that *PbrCAMTA2* was overexpressed over 23-fold in pear fruits compared to the tissues injected with the empty vector (Figure 8C). We also observed that the expressions of *TBG*, *XET*, and *ACO* were lower in fruits with overexpressed *PbrCAMTA2*, indicating that *PbrCAMTA*2 may delay pear fruit ripening by regulating *TBG*, *XET*, and *ACO*.

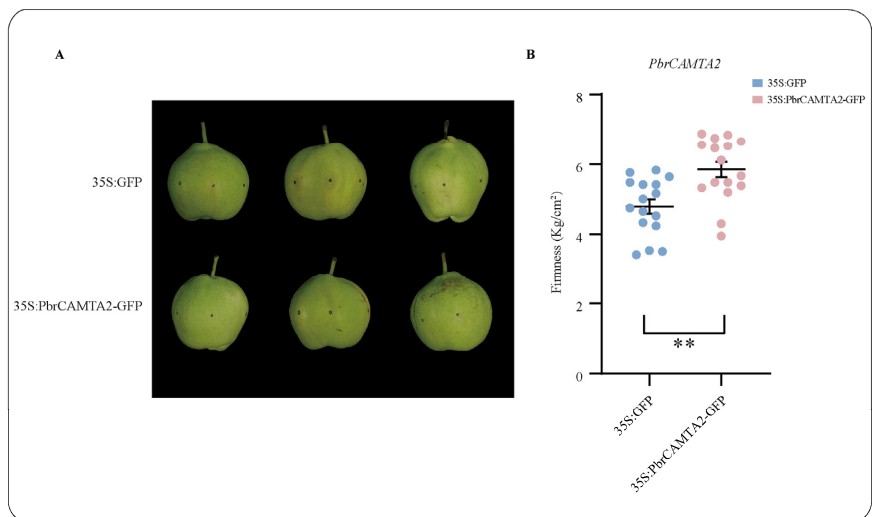

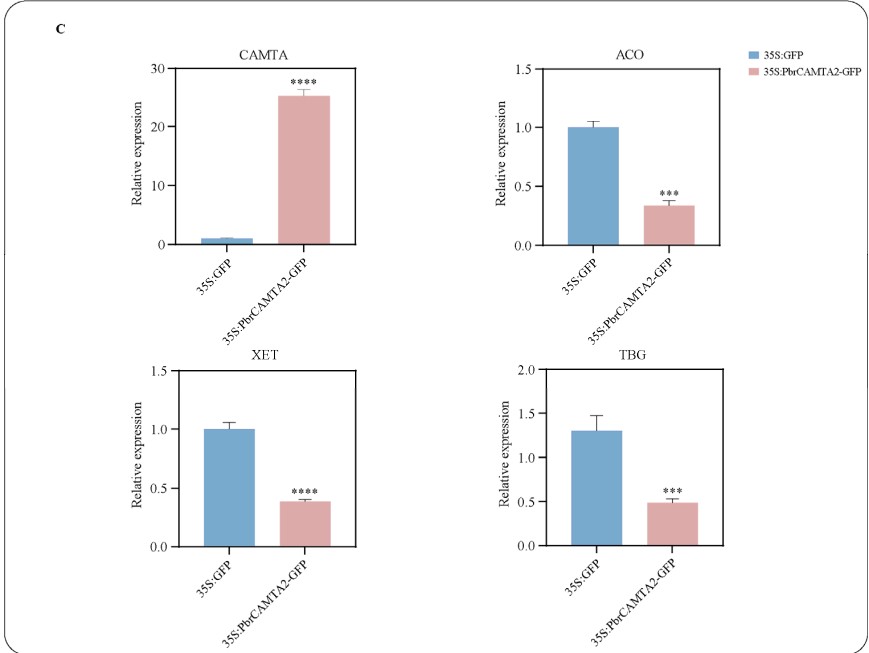

**Figure 8.** Transient assays by overexpressing *PbrCAMTA2* in 'Zaosu' pear fruit 110 DAFB. (**A**) Phenotypes

of pear fruits. Images were taken five days after agro-infiltration. (**B**) The firmness of fruits. The mean separations are conducted by *t*-test in the GraphPad Prism 8. Asterisks indicate that significant differences existed (** *p* < 0.01, *** *p* < 0.001 and **** *p* < 0.0001). (**C**) qRT-PCR analysis of *PbrCAMTA*, *ACO*, *XET*, and *TBG* genes in different experimental groups. The pink columns represent *PbrCAMTA2*, and the blue columns represent the empty vector. Error bars indicate three technical replicates derived from one bulked biological replicate.

## 4. Discussion

### 4.1. Identification and Phylogenetic Analysis of the CAMTA Family in Eight Rosaceae Species

We identified a total of 46 CAMTA proteins from eight species of the Rosaceae family, nine of which were in Chinese white pear, nine in European pear, eight in apple, four in black raspberry, three in strawberry, four in Japanese apricot, four in sweet cherry, and five in peach. Additionally, two model plants were introduced to assist in the *CAMTA* genes classification. Finally, the 46 CAMTA proteins were clustered into five subgroups (Figures 1 and 2). Our results are similar to the clustering results obtained for other species, as ten *VvCAMTA* genes are classified into four groups in grapes [11], and nine *CitCAMTA* genes are clustered into five groups in citrus [16]. The distribution of the proteins from each Rosaceae species was various. Proteins in the same subgroup often have similar functions. In addition, based on the results of the multi-sequence comparison and conservative domain analysis, 98.30% of the CAMTA members had a conserved DNA-binding domain (CG-1), consisting of approximately 130 amino acids, which is involved in regulating the transcription and expression of genes associated with ethylene, abscisic acid, and the light-responsive element [32]. All of the *CAMTA* genes had a transcription factor immunoglobulin domain (TIG), which is involved in the non-specific binding of various transcription factors, initiating the dimerization of proteins [10]. CAMTA proteins of different subgroups all contained those two motifs, suggesting that the Rosaceae CAMTA family is a highly conserved transcription factor family.

In addition, gene structure analysis showed most *CAMTA* genes (87%) contained 9–13 introns (Figure 3), similar to the number of introns in *CAMTA* genes in other species: 10–13 in wheat [33], 9–12 in *Gossypium* [15], 6–12 in citrus [16], and 10–12 in banana [17]. We observed the physical properties of the CAMTA protein sequences of eight Rosaceae species (Supplementary File S6: Table S2); the results showed that the CAMTA protein family generally had longer sequences (854–2065 amino acids) and larger protein molecular masses (96–233 kD), much higher than the other gene families in Rosaceae, such as the ADH protein family, which has a length ranging from 300 to 887 amino acids and an MW ranging from 32.32 to 69.83 kD [34].

### 4.2. Evolutionary Mechanism and Good Collinearity Relationships between Pear and Other Seven Species of the Rosaceae Family

Gene duplication events can be mainly traced to five origins: genome-wide duplication (WGD), tandem (TD), proximal (PD), transposed (DD), and dispersed duplications (DSDs) [35]. Many gene families have expansion events in Rosaceae [36]. Previous research has shown that there are two whole-genome duplication (WGD) events that occur in pear and apple, with a recent WGD event taking place at 30–45 MYA and an ancient WGD event that occurred at ~140 MYA [23,37]. In this research, eight of the nine *PbrCAMTA* genes in pear resulted from WGD/segmental duplication, and one remaining *PbrCAMTA* gene resulted from dispersed duplication (Supplementary File S8: Table S4), indicating that the WGD/segmental event was the main driving force in the evolution of *CAMTA* genes in Chinese white pear. Researchers have also proved that a similar situation occurred in other gene families in pear. The *SWEET* gene family, for instance, is found to have WGD/segmental duplication as the primary evolutionary mechanism [38].

Between Chinese white pear and the other seven Rosaceae species, we detected good collinearity relationships (Figure 4A). We found that the most collinear gene pairs (14) were

shared between Chinese white pear and apple. Considering that both pear and apple have undergone a recent WGD event, they are expected to have a closer relationship. Only ten collinear pairs shared between Chinese white pear and European pear may be due to the interspecific divergence that arose during evolution. Among the nine *PbrCAMTA* genes, *PbrCAMTA6* had the most collinear gene pairs (13) across seven species, indicating that *PbrCAMT6* may have evolved from ancestral pear genes. In addition, *PbrCAMTA3* shared only five collinear gene pairs with the other seven species, probably due to the dispersed duplication event that caused conserved domain variation during evolution.

### 4.3. Cis-Acting Element Analysis of CAMTA Genes

Zein is a natural storage protein, and researchers have discovered that zein coatings can delay the ripening process of mango by controlling the activity of softening enzymes, such as pectin methyl esterase and cellulase [28]. In pears, we found that 55.56% of *PbrCAMTAs* had zein metabolism-regulation-responsive elements in their promoters in Chinese white pear, while only 11.11% of *PcpCAMTAs* had the element in *P. communis*. There are different maturity phenotypes between the two pear cultivars; the fruits of *P. communis* have a post-ripening process and usually have a good flavor after post-harvest storage, while *P. bretschneideri* fruits can be eaten immediately after harvest [25]. Based on this, we speculated that the zein metabolism-regulation-responsive element regulates pear fruit softening divergence between Chinese white pear and European pear cultivars by participating in differential expressions of *CAMTA* genes.

Most of the *CAMTA*-promoter cis-acting elements of Rosaceae species contained ABA-responsive elements, especially in pears, with the ABRE element being present in all *PbrCAMTA* genes. The ABRE element is known to act as a signal response cis-acting element to conduct $Ca^{2+}$ signals, and CAMTA transcription factors in pear may be associated with responses to ABA and environmental stress [39]. *PbrCAMTAs* may participate in the Abscisic acid (ABA) metabolic pathway through the ABA cis-acting element in order to regulate pear fruit ripening. In addition, previous studies have found that hormones, such as Gibberellins (GA) and Methyl Jasmonate (MeJA), are related to fruit firmness. Gibberellin could delay maturation by upregulating auxin [40], while MeJA could accelerate fruit softening by promoting the expression of *XTH1* and *EG1* [41].

### 4.4. Expression Analysis and Functional Prediction of Candidate PbrCAMTA Genes

*CAMTA* family members have proven to be involved in fruit softening, signal transduction, and abiotic and biotic stress response during the development of many plants [8,12,16,17,42]. However, there have only been a few studies on CAMTA family members in relation to pear. Our research on *PbrCAMTAs* expression patterns has the potential to provide solid bases for further analysis. In transcriptome analysis, nine *PbrCAMTA* genes showed diverse expression patterns in different tissues and during different fruit development stages (Figures 6A and 7A). From the results of qRT-PCR (Figures 6B and 7B), we observed that during the fruit development of 'Dangshansuli' pear, the expression of most of the *PbrCAMTA* genes was higher before fruit development 36 DAFB, indicating that *PbrCAMTAs* may affect the development of young fruits. During 80 DAFB-167 DAFB, the expression of *PbrCAMTAs* showed an increasing trend, and a similar trend has been detected in tomatoes [8]. *PbrCAMTAs* are likely to be involved in fruit development and softening.

We found that most *PbrCAMTA* genes were expressed at high levels in leaves, suggesting those *PbrCAMTA* genes may play crucial roles in pear leaf tissues and plant stress response [9,43]. Proteins in the same subgroup often have similar functions, which might be traced back to a common ancestral protein [36]. *PbrCAMTA8* was classified in Group V, together with *AtCAMTA3*, which has been reported to negatively regulate salicylic acid-mediated plant immunity [22], suggesting that the *PbrCAMTA8* gene might function similarly to *AtCAMTA3* and could be involved in SA metabolism, negatively regulating immunity in pear. Fruit setting is the developmental transition from the ovary to the young fruit [44]. We found that *PbrCAMTA2* had a higher expression level in the ovary.

*PbrCAMTA2* may be related to fruit setting by affecting the development of the ovary and that of the young fruit. Additionally, we noticed that *PbrCAMTA2* had a trend of high expression in the early stages of development and low expression in the ripening stage. Previous studies have shown that CAMTA is involved in fruit softening [8]. Therefore, we speculated that *PbrCAMTA2* might be involved in the softening of pear fruits. This conjecture was confirmed by transient transformation results (Figure 8B,C). It is believed that fruit softening is due to ethylene-mediated enzymatic modifications of the cell wall. Different types of cell-wall-modifying enzymes, such as TBG and XET, are thought to play important roles in the ripening process [5,45]. ACO is believed to participate in the de novo synthesis of ethylene [7]. Based on these findings, we chose to consider these genes for further research on fruit softening. Finally, our results indicated that overexpressing *PbrCAMTA2* caused significant decreases in the expression levels of cell-wall-related genes: *XET*, *TBG*, and ethylene-synthesis-related genes, such as *ACO* (Figure 8C). We performed a cis-acting analysis of *PbrACO*, *PbrXET*, and *PbrTBG* putative promoters (2000 bp upstream) using the PlantCARE website in order to explore possible relationships with *PbrCAMTA2* further. As shown in Figure S4 (Supplementary File S4), ABRE elements were widely found in the promoter regions (2000 bp) of three genes. It has been reported that as the $Ca^{2+}$-dependent transcription factor in plants, CAMTA may function as a link between $Ca^{2+}$ signaling and ABRE-related cis-elements [46,47]. Therefore, we speculated that *PbrCAMTA2* regulates downstream gene expression, such as softening-related enzymes (TBG, ACO, and XET), by binding to ABRE elements. Therefore, we hypothesized that *PbrCAMTA2* might influence pear fruit ripening and extend the storage period of pears.

## 5. Conclusions

We identified 46 *CAMTA* genes in the genomes of eight Rosaceae species and studied their physicochemical properties. We clustered 59 CAMTA proteins from the eight species and two plant model species into five subgroups. Duplication events analysis proved that WGD/segmental duplication and dispersed duplication were the main driving force in the expansion of the *PbrCAMTA* gene family. We identified four homologous WGD gene pairs in the *PbrCAMTA* family. The promoters of nine *PbrCAMTA* genes contained cis-acting elements related to light response, hormone response, and stress response. Through transcriptome data and qRT-PCR experiments, we concluded that the expression of *PbrCAMTA* is different in plant growth and in development. Transient transformation assays demonstrated that *PbrCAMTA2* could regulate fruit softening. Our results help to clarify the biological function of *PbrCAMTA* genes in pear development (pear fruit development and different tissues) and have a significant influence on our knowledge of woody plant *CAMTAs*.

**Supplementary Materials:** The following supporting information can be downloaded at: https://www.mdpi.com/article/10.3390/horticulturae9040467/s1, Figure S1: Amino acid compositions of 20 conserved motifs of the CAMTA family in eight Rosaceae species; Figure S2: Multiple sequence alignment of CAMTA protein family in ten species; Figure S3: Chromosomal distribution and Synteny analysis of *CAMTA* genes; Figure S4: Cis-acting analysis of *PbrACO*, *PbrXET*, and *PbrTBG* putative promoters (2000 bp upstream); Table S1: Primers chart for qRT-PCR and vector construction; Table S2: Characteristics of *CAMTA* genes in eight Rosaceae species; Table S3: The number of CAMTA of eight Rosaceae species in five subgroups of phylogenetic tree; Table S4: Duplication events types of *PbrCAMTA* gene family; Table S5: Cis-acting elements heatmap in the putative promoter of *CAMTA* genes in eight Rosaceae species.

**Author Contributions:** Conceptualization, J.Y. and B.S.; methodology, J.Y., B.S. and B.C.; software, J.Y. and B.S.; validation, J.Y. and K.G.; formal analysis, J.Y. and K.G.; investigation, B.C.; data curation, K.Z.; writing—original draft preparation, J.Y.; writing—review and editing, B.S. and J.L.; funding acquisition, J.W. All authors have read and agreed to the published version of the manuscript.

**Funding:** This work was funded by the National Key Research and Development Program of China (2021YFD1200200), the National Natural Science Foundation of China (31725024 and 31801835), the Earmarked Fund for China Agriculture Research System (CARS-28), and the Earmarked Fund for Jiangsu Agricultural Industry Technology System (JATS [2021]453).

**Institutional Review Board Statement:** Not applicable.

**Informed Consent Statement:** Not applicable.

**Data Availability Statement:** The datasets generated for this study are available upon request to the corresponding author.

**Conflicts of Interest:** The authors declare no conflict of interest.

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
