# Peer review of "Genome-Wide Identification and Expression Analysis of CAMTA Gene Family Implies PbrCAMTA2 Involved in Fruit Softening in Pear"

_horticulturae, doi:10.3390/horticulturae9040467_

Round 1

Reviewer 1 Report

The work of Yu et al. makes a clear and expanded analysis of CAMTA genes in Chinese White pear (Pyrus bretschneideri) and other 7 Rosaceae species. This approach have become a common approach for recent sequenced species and provides a new layer of information based on previously published genome data. Interestingly, this work provides additional experimental information about the functional role of PbrCAMTA2 on Chinese White pear softening using an ectopic expression assay on fruit and analyzing the infiltrated tissue looking for firmness changes and the expression of softening marker genes. Based on authors description of results and I cite: "We also observed that the expression of TBG, XET and ACO were lower in fruits with overex pressed PbrCAMTA2, indicating PbrCAMTA2 may delay pear fruit ripening by regulating TBG, XET and ACO."; I would suggest to authors that perform a promoter analysis of TBG, XET and ACO genes looking for the DNA binding motif for CAMTA transcription factors. This analysis could improve the interpretation and implicances of the effect of the overexpression of PbrCAMTA2 on TBG, XET and ACO transcript levels in Chinese White pear fruits.

Author Response

Reviewer reports:

Reviews 1

1、The work of Yu et al. makes a clear and expanded analysis of CAMTA genes in Chinese White pear (Pyrus bretschneideri) and other 7 Rosaceae species. This approach has become a common approach for recent sequenced species and provides a new layer of information based on previously published genome data. Interestingly, this work provides additional experimental information about the functional role of PbrCAMTA2 on Chinese White pear softening using an ectopic expression assay on fruit and analyzing the infiltrated tissue looking for firmness changes and the expression of softening marker genes.

Author response: We greatly thank for the positive comments for the importance and value of our study, and greatly appreciate constructive comments and suggestions, which help to improve the quality of manuscript. We have revised the manuscript according to the suggestion and addressed question point by point carefully as following responses to reviewers.

2、Based on authors description of results and I cite: "We also observed that the expression of TBG, XET and ACO were lower in fruits with over expressed PbrCAMTA2, indicating PbrCAMTA2 may delay pear fruit ripening by regulating TBG, XET and ACO."; I would suggest to authors that perform a promoter analysis of TBG, XET and ACO genes looking for the DNA binding motif for CAMTA transcription factors. This analysis could improve the interpretation and implicates of the effect of the overexpression of PbrCAMTA2 on TBG, XET and ACO transcript levels in Chinese White pear fruits.

Author response: Thank you for your suggestion. We performed cis-acting element of PbrACO, PbrXET and PbrTBG putative promoters (200 bp upstream) using PlantCARE website in order to explore possible regulation relationships with PbrCAMTA2. As shown in the Figure R1, ABRE elements were widely found in the promoter regions (2000 bp) of three genes. It has been reported that as the Ca2+-dependent transcription factor in plant, CAMTA may function as a link between Ca2+ signaling and ABRE-related cis‑elements (Finkler et al., 2007; Galon et al., 2010). Therefore, we speculated that PbrCAMTA2 might regulate downstream gene expression such as softening-related enzymes (TBG, ACO and XET) by binding to ABRE elements. We have added more detail description in this revision, please see the Line 482-490 (Line 509-516 under the “Track Changes”) in revised version.

Figure R1: Cis-acting analysis of PbrACO, PbrXET and PbrTBG gene putative promoter regions (2000 bp upstream)

Reference:

  1. Finkler A, Kaplan B, Fromm H. Ca-Responsive cis-Elements in Plants[J]. Plant Signaling & Behavior, 2007, 2: 17-19.
  2. Galon Y, Finkler A, Fromm H. Calcium-regulated transcription in plants[J]. Molecular Plant, 2010, 3: 653-669.

Reviewer 2 Report

Dear authors,

I like your paper. Nevertheless, it has some problems with English. The common one is plural/singular usage, another one is word order. You say a gene is located to nucleus, indeed, it should be the gene product, i.e. the protein. Some other problems, small or large, are pointed at below.

Line 11 plural is better

line 17 the not a

line 21 genes or proteins?

Line 147 35S:PbrCAMTA::GFP vectors Use :: appropriately.

Line 314 rewrite the definition of RPKM

You did qRT-PCR. What makes you think the signal comes from amplification of CAMTA RNA, and not from some other stuff?

What tissues is Fig 7B legend about? Please correct.

Results 3.7 produces more questions than answers. It’s no secret that transcription factors are localized to nucleus. Does tobacco work as pear? GFP itself is known (and shown here, too) to like to go to nucleus. I think 3.7 has no merit and should be removed.

When you correct the above and the like issues, the paper will be OK.

Author Response

Reviewer reports:

Reviewers 2

Dear authors,

1、I like your paper. Nevertheless, it has some problems with English. The common one is plural/singular usage, another one is word order. You say a gene is located to nucleus, indeed, it should be the gene product, i.e. the protein. Some other problems, small or large, are pointed at below.

Author response: We greatly appreciate for the positive comments and suggestions from the reviewer, which will help to improve the quality of manuscript. We have revised the manuscript according to the suggestions and addressed each question point by point carefully as following responses to reviewers.

2、Line 11 plural is better

Author response: Thank you for your nice suggestion. We have revised in Line 11 (Line 11 under the “Track Changes”).

3、Line 17 the not a

Author response: Sorry for the unclear description. We have revised in Line 17 (Line 17 under the “Track Changes”).

4、line 21 genes or proteins?

Author response: Thank you for your suggestion. The sentence has been deleted in the Line 20 since we removed the subcellular localization section.

5、Line 147 35S:PbrCAMTA::GFP vectors Use :: appropriately.

Author response: Thank you for your suggestion. We have checked and revised the names with “p1300-35S:PbrCAMTA2-GFP”. Please see the Line 147 and 351 (Line 158-159 and 376 under the “Track Changes”) in this revision.

6、Line 314 rewrite the definition of RPKM

Author response: Thank you for your correction. We have checked and revised the definition of RPKM with “Reads Per Kilobase Per Million Mapped Reads”. Please see the Line 132-133 (Line 134-135 under the “Track Changes”) in this revision.

7、You did qRT-PCR. What makes you think the signal comes from amplification of CAMTA RNA, and not from some other stuff?

Author response: Thank you for your nice suggestion. Before the experiment, primers were identified specifically using internal website (http://202.195.250.6:8080/src/#/home) of our library. As shown in the Figure R1, the reverse primer was only corresponded to Pbr035644.2 (PbrCAMTA2), therefore, this pair of primers of PbrCAMTA2 was considered specifically. Once the qRT-PCR experiment end, we analyzed the primer melting curves and there was only one melting peak (Figure R2). In summary, the signal was come from RNA amplification of PbrCAMTA2.

Figure R1: PbrCAMTA2 primer-specific identification for qRT-PCR

Figure R2: Melting curve of PbrCAMTA2 in qRT-PCR

8、What tissues is Fig 7B legend about? Please correct.

Author response: Sorry for the unclear description. The fruit tissues at different DAFB of ‘Dangshansuli’ were used to perform further qRT-PCR experiments. We have added the relevant description in the Line 137 and 343 (Line139 and 355 under the “Track Changes”).

9、Results 3.7 produces more questions than answers. It’s no secret that transcription factors are localized to nucleus. Does tobacco work as pear? GFP itself is known (and shown here, too) to like to go to nucleus. I think 3.7 has no merit and should be removed.

Author response: Thank you for your valuable suggestion. We agreed the reviewer’s opinion and removed the section of Results 3.7 in this version.

10、When you correct the above and the like issues, the paper will be OK.

Author response: Thank you for your kind and valuable review. In this revision, we have given full consideration to the suggestions of the reviewer and corrected and addressed each question point by point carefully to reviewers.

Reviewer 3 Report

Congratulations to the authors for this impeccably written paper! I appreciate the fact that work has been done on a sufficiently large number of species, varieties, organs, tissues and phenological phases to be able to draw credible conclusions regarding the expression analysis of CAMTA gene family.

However, I ask the authors to explain and elaborate, how can the results of this research be used in plants breeding, for the early elimination from experimental fields of pears with low firmness?

In terms of keywords, in my opinion, the word "evolution" should be replaced with a more relevant word for this paper.

Author Response

Reviewer reports:

Reviewers 3

1、Congratulations to the authors for this impeccably written paper! I appreciate the fact that work has been done on a sufficiently large number of species, varieties, organs, tissues and phenological phases to be able to draw credible conclusions regarding the expression analysis of CAMTA gene family.

Author response: We greatly appreciate the review’s positive attitude and constructive comments for our manuscript, which help to improve the quality of the manuscript. We have revised the manuscript according to the suggestions, and addressed each question point by point carefully as following responses to reviewers.

2、However, I ask the authors to explain and elaborate, how can the results of this research be used in plants breeding, for the early elimination from experimental fields of pears with low firmness?

Author response: Thank you for your nice suggestion. Commercially, postharvest fruits are often hard to store, and fruit quality is reduced. It is of great significance to carry out research on pear fruit softening. With the changes of the market demand and natural environment, breeding is very important to meet the consumer demands. Molecular breeding technologies such as transgenic technology and molecular marker technology have the advantages of short cycle, high efficiency, and high precision in directional breeding. Applying those technologies to pear breeding could greatly improve the quality of pear fruit and enhance industrial competitiveness (Khlestkina and Shavrukov, 2022). Unfortunately, this article did not distinguish between high or low pear fruit firmness using molecular markers, but functional characterization by transient transgenesis of PbrCAMTA2 gene was hoped to lay a foundation for subsequent breeding research in pear.

Reference:

  1. Khlestkina E, Shavrukov Y. Molecular-Genetic Basis of Plant Breeding. Biomolecules, 2022, 12: 1392.

3、In terms of keywords, in my opinion, the word "evolution" should be replaced with a more relevant word for this paper.

Author response: Sorry for the unclear description. We have corrected it with “Synteny analysis” in the Line 25 (Line 27 under the “Track Changes”).

.
